# Research Progress on Blue-Phase Liquid Crystals for Pattern Replication Applications

**DOI:** 10.3390/ma16010194

**Published:** 2022-12-26

**Authors:** Hao Wang, Huimin Zhou, Wanli He, Zhou Yang, Hui Cao, Dong Wang, Yuzhan Li

**Affiliations:** School of Materials Science and Engineering, University of Science and Technology Beijing, Beijing 100083, China

**Keywords:** blue phase, patterned, micropattern, electrically responsive, handwriting, mask, inkjet printing

## Abstract

Blue-Phase Liquid Crystals (BPLCs) are considered to be excellent 3D photonic crystals and have attracted a great deal of attention due to their great potential for advanced applications in a wide range of fields including self-assembling tunable photonic crystals and fast-response displays. BPLCs exhibit promise in patterned applications due to their sub-millisecond response time, three-dimensional cubic structure, macroscopic optical isotropy and high contrast ratio. The diversity of patterned applications developed based on BPLCs has attracted much attention. This paper focuses on the latest advances in blue-phase (BP) materials, including applications in patterned microscopy, electric field driving, handwriting driving, optical writing and inkjet printing. The paper concludes with future challenges and opportunities for BP materials, providing important insights into the subsequent development of BP.

## 1. Introduction

Blue-phase liquid crystals (BPLCs) [1,2,3] are fascinating self-assembling 3D nanomaterials. As early as 1888, scientists measuring the liquid crystal phase behaviors of cholesteryl benzoate found that [4]: “a bright blue-purple phenomenon appeared during the cooling of the melted compound, and this bright blue-purple, which soon disappeared, was followed by a cloudy phenomenon. The same color effect reappeared again when the cooling was continued. At the same time, the sample began to crystallize and the color disappeared”. This color is an optically isotropic result, indicating a thermodynamically stable state. It is called a blue phase because of its blue Bragg reflection. At present, research on BPLCs is mainly focused on the display field, such as broadening the blue-phase temperature range [1,2,5,6,7,8,9,10,11] and realizing blue-phase liquid-crystal displays [12,13,14,15]. Nevertheless, significant progress has been made in its non-display areas, such as large-domain films [3,16,17], blue-phase lasers [18,19,20], etc.

Patterned applications have excellent prospects in the field of information transmission and display [21,22,23,24,25,26,27,28,29]. The use of liquid crystals in patterned applications has always been a focus of attention, and the most well-known products of patterned applications based on liquid crystals are various kinds of liquid-crystal displays, electronic paper, etc., which are ubiquitous in people’s lives. A variety of responsive patterned applications have been developed based on liquid crystals [30,31,32,33,34,35,36]. However, the current challenges of liquid-crystal display (LCD) patterned applications include long response times, poor contrast, poor field of view, single color, complex preparation processes and high dependence on orientation layers. Polymer cholesteric liquid-crystal (PCLC) sheets have the potential to be used in many passive and active optoelectronic applications, including military vehicles, smart windows, color filters, multi-color flexible displays, etc. [37,38,39,40,41,42,43]. This offers the possibility of thin, light, reflective, lightweight and flexible devices with low power. However, PCLC suffers from long response times, thermal ageing properties and poor mechanical properties. Significant research has been undertaken to improve the performance of liquid-crystal devices, such as doping with nanoparticles [44,45,46,47,48]. Recently, blue-phase liquid crystals with optically isotropic [49], fast Kerr-effect [50] and three-dimensional cubic structures have been widely used for patterned displays. The schematic diagram of the fast Kerr effect for blue phase liquid crystals is shown in Figure 1. Blue-phase-based patterned applications have the advantages of sub-millisecond response time [51], simple preparation process, high contrast ratio and a wide range of applications. Therefore, the pattern reproduction properties of blue phase liquid crystal have important application value in information transmission, display, handwritten paper, high-resolution graphics and so on. The aim of this review is to present the latest important advances in the field of patterned BPLC materials. This review introduces applications of micro-patterns, electric field driving, handwritten paper, photo-mask pattern preparation and inkjet printing of patterns, discusses the limitations of applications in pattern display and microelectronics, and finally summarizes the opportunities and challenges of blue-phase pattern replication in advanced functional material design and device applications.

## 2. Patterned Applications of Blue Phase Liquid Crystals

### 2.1. Application of Blue-Phase Spontaneous Micropattern

Different phases of liquid crystals show different textures and patterns under the polarizing microscope. For example, in cholesteric-phase liquid crystals, distorted grain boundary phases show interesting patterns, whereas BPLC shows regular mosaic patterns under a polarizing microscope. BP is a thermodynamically stable phase with a three-dimensional lattice structure [52,53]. The phase transition of BPLC has now been characterized using polarization optics [54,55,56]. The arrangement of molecules in blue-phase liquid crystals tends to twist not only in the direction of the spiral axis, but also in a direction perpendicular to the spiral axis, which is called a biaxial helical structure [57]. Dopants with medium torsional power induce the cholesteric phase when the pointing vector twists along a single axis, whereas strong torsional dopants induce the pointing vector to twist in all directions perpendicular to itself, resulting in the so-called double-twisted cylinder (DTC) [58]. In BPLC, the liquid crystal molecules are first arranged in a double-twisted arrangement to form DTCs, and then the lattice structure of BPLC is constructed by supramolecular self-assembly, which inevitably forms defects in the three-dimensional long-range space. In the three-dimensional space, the defects appear at regular intervals to form a cubic lattice, just like the lattice of a solid crystal. In other words, BP is a thermodynamically stable phase that coexists with defects. In terms of crystal structure, BP can be divided into three sub-phases, corresponding to BPI, BPII and BPIII, whose corresponding lattice structures are the body-centered cubic structure (the space group of defects is I4_1_32), the simple cubic structure (the space group of defects is space group P4_2_32) and the amorphous state, respectively. As the temperature increases, the sequence of Ch (cholesteric phase)→BPI→BPII→BPIII→I (isotropic phase) appears [59]. The theoretical model of the blue phase is shown in Figure 2. To date, various optical devices have been manufactured based on the BPLC phase-change process [60,61]. Jiang et al. used the non-diffusive phase transition characteristics of blue-phase liquid crystals to prepare large color-block multi-domain BPLC films to achieve micro-area laser and temperature tunable binary/trinary QR codes in 60 μm crystal domains [62]. As shown in Figure 3, a binary code is obtained by grayscale conversion, binarization, pixelization and encoding of the BP, and the temperature is switched to achieve a rapidly switchable 2D code. This switchable QR code can be used as an anti-counterfeit ‘ID’ card for medicines, fine wines, watches and jewelry, as it offers a higher level of security than a normal static QR code. However, the validation process for such films is complex and needs to be carried out under specific conditions. In addition, how to maintain stability over a long period of time is a problem that needs to be solved.

### 2.2. Electrically Responsive Blue-Phase Patterned Mode–BPLCD

Optically isotropic blue-phase liquid crystals exhibit the Kerr effect in the presence of an electric field and therefore hold good promise for use in liquid-crystal display materials. Blue-phase liquid-crystal display (BPLCD), which presents patterns through electric field control, has the advantage of fast response, not only enabling LCDs to achieve field-sequential color display mode, but also greatly reducing dynamic artefacts, and optimizes resolution and optical efficiency, and therefore is considered to be the basis of the next generation of LCDs and is being studied by a wide range of scholars. For example, in 2008, Samsung demonstrated the world’s first BPLCD at the SID (the Society for Information Display), as shown in Figure 4 [63]. BPLCDs mainly include in-plane-switching [64,65,66,67] and vertical-field-switching [68,69] electrode structures. BPLCDs are isotropic in the absence of an electric field, so they can be prepared without an orientation layer and have extremely high contrast and viewing angles. Because of the fast response of the blue phase itself, BPLCDs can continue to be driven using RGB color-timing technology, which not only saves costs but also increases transmittance and resolution. However, the drive voltage of BPLCDs with in-plane-switching electrode structures is very high (> 20 V). Although it can be reduced by means of raised electrode structures, double penetration electrode structures or wall electrode structures, these methods inevitably introduce other problems such as process complexity and reduced transmittance. BPLCDs with a vertical-field-switching electrode structure also have problems such as poor contrast and visual angle. Although this can be improved by changing the polarization angle, they still fail to meet the practical requirements. Improving the performance of BPLCD is a challenge worthy of further study.

Recently, reflective BPLCDs have attracted interest due to their good contrast in bright light and conservation of energy. Unlike transmissive BPLCDs, where the light source is placed under the lower substrate, reflective BPLCDs have the light source on the upper substrate and the incident light enters the liquid-crystal cassette from the upper substrate, where the incident light can accumulate double the optical range difference in the liquid-crystal layer, thus allowing for a lower operating voltage. In 2013, Yan et al. showed a full-color reflective blue-phase liquid-crystal display with polymer-stabilized red, green and blue sub-pixels via electric field induction [70]. Due to the characteristic polarization of selective reflection, the proposed reflective display showed potential applications in 3D displays, where 3D images can be viewed through circularly polarized displays. In order to further improve the driving voltage of polymer-stabilized blue-phase reflective displays, Luo et al. achieved low driving voltages by doping a small amount of ferroelectric nanoparticles (BaTiO_3_) [71]. As shown in Figure 5, compared with polymer-stabilized blue-phase liquid crystal (PSBPLC) without ferroelectric nanoparticles, the vertical driving electric field of PSBPLC with 0.4 wt% BaTiO_3_ ferroelectric nanoparticles is significantly reduced from 6−7 V/um to 1.8 V/um for red, green and blue cells, a significant reduction of more than 70%. This greatly enhances the practical prospects for blue-phase reflective displays.

At the same time, Luo et al. developed a super-reflective, electrically switchable, fast-responding and color-reflective display based on multilayer BPLC films to solve the problem of reflective displays with single-layer BPLC films having reflectance below 50% in the visible range [72]. As shown in Figure 6, by filling a multilayer blue-phase liquid-crystal film with an independent porous polymer network with an achiral nematic liquid crystal, high reflectivity of 89%, 82% and 68% is achieved in the red, green and blue reflection color regions, respectively. This electrically switched super-reflective BPLC film with sub-millisecond response times significantly improves the performance of high-reflectance color reflective displays. The technology can also be used for switchable optoelectronic devices, lasers, mirrors, etc.

In contrast to the Kerr effect, the shift in Bragg reflection wavelengths caused by electric fields, namely electrostriction, has received little attention due to the short range of switching wavelength changes. Polymer networks used to stabilize BPLC may hinder lattice changes under electric fields [1]. Lu reported the first pioneering work on electrically switchable color reflections in PSBPLC [73]. As a result of the onset of electrostriction and phase transition from BPII to BPI, an extensive but discontinuous movement of photonic bandgap (PBG) is observed, as shown in Figure 7. Compared with polymer-stabilized cholesteric liquid crystals [74], PSBPLC offers a narrower half bandwidth and less than half the drive voltage required to electrically switch color. This discovery provides new ideas for electrogenic BP patterns. Lin reported on the electrodynamic displacement and swelling of PBG in polymer-stabilized blue-phase systems under DC fields [75]. However, research in this area is still at a preliminary stage and further studies are needed to elucidate the underlying mechanisms and improve performance.

BPLCDs are considered to be the next generation of LCDs due to their fast response time and low cost. However, transmissive BPLCDs still suffer from high operating voltage, poor contrast, poor viewing angle and the hysteresis effect. Reflective BPLCDs have improved reflectivity and operating voltage, but the process is complex and the operating voltage is still not up to the requirements. Future research should pay more attention to the innovation of the device structure, and propose a new structure that can improve the electro-optic properties of BPLCDs and is easy to mass-produce. At the same time, BPLCDs with a larger Kerr coefficient, wider temperature range and smaller pyroelectric effect can be obtained by doping suitable nanomaterials.

### 2.3. Blue-Phase Patterned Mode Based on Handwriting

Rewritable display materials, which repeatedly perform a ‘write-erase’ cycle by switching color, are a potential alternative to traditional paper and could help to alleviate global deforestation, which is mainly caused by increased paper consumption. As one of the color-rendering materials, responsive photonic crystals consist of periodic structural materials with different refractive indices that exhibit tunable structural color [76,77,78]. At present, a variety of handwriting films have been developed based on liquid crystals, but these films have problems such as handwriting divergence and poor contrast [34,79]. BP-based handwriting display devices have the advantages of high contrast ratio, simple preparation and good stability.

Among the currently available nanomaterials, magnetic (Fe_3_O_4_) nanoparticles have attracted significant attention in terms of technological applications [80,81,82,83,84,85], especially in the form of ferromagnetic fluids. He et al. prepared a novel powerless magnetically driven LC flexible display using BPLC doped with magnetic Fe_3_O_4_ nanoparticles [86]. As shown in Figure 8, the magnetic pen writes on the outer surface of the BPLC cell and the magnetic nanoparticles are attracted to the inner surface of the cell, thus allowing clear handwriting as well as patterns to be obtained, relying on the same principle to enable the erasure of handwriting. This composite material is inexpensive, simple to prepare, and has clear text. Moreover, it is easily erasable and is environmentally friendly and therefore can be used to replace traditional displays in the classroom.

Photonic shape-memory polymers are advanced polymers that have a shape-memory effect accompanied by a color change; they can block the propagation of light under certain conditions [87,88,89]. Yang et al. reported a fabricated freestanding BP film of photonic shape-memory polymers achieving high optical reversibility for patterned reconfigurable BP films [90]. Such photonic films have excellent optical properties [91]. As shown in Figure 9, a blue “BP” pattern was written on the green BP film using a shape-memory programming process with specific pressure on the film, which could be erased by a shape-memory recovery process after the polymer film was heated. The pressure-driven handwritten film is simple to prepare and easy to eliminate, and is expected to be an alternative to traditional paper.

Recently, Yang et al. achieved reprogrammability, reconfigurability and visualization by using an elaborate system consisting of specially designed hydrogen-bonded mesocrystalline precursors to fabricate freestanding blue-phase liquid-crystal films that display reversible humidity-responsive behavior by manipulating the lattice parameters of their nanostructures [92]. As shown in Figure 10, writing through a stylus containing water, the BPLC film rapidly changes from blue to red, then due to its hydrophilicity, rapidly changes to green and finally back to red again. The bold “BP” is written in the film with water and then spontaneously erased when the water evaporates. Subsequently, the fine-type “LC” is obtained by writing a second time. The response behavior of this humidity-responsive BPLC film is reprogrammable, reconfigurable and can be easily visualized.

Handwriting-driven BP-patterned material is a reliable alternative to conventional paper and can be reused by means of “write-erase”. This easily prepared and low-cost display material has great potential in the field of handwriting paper. However, the resolution of BPLC film pattern driven by handwriting is low, and thus it cannot achieve fine patterns, and the high-resolution patterning display requires an optical mask and inkjet printing.

### 2.4. Mask-Based Optical Writing of Blue-Phase Patterned mode

In recent years, mask-based BP patterned optical writing mode has attracted people’s attention due to its optical rewritability and high contrast ratio. This mode simultaneously solves the problems of the complex process flow and short retention time of traditional LC-based optical writing patterned devices [93,94,95,96,97,98,99].

Light-driven self-organized BP 3D-cubic nanostructures displaying unique photon reflections in three orthogonal directions have received increasing attention, but their reflection wavelength tuning is usually very narrow [100,101,102]. In some recent reports, a uniformly aligned BPII structural domain was obtained by friction treatment of the substrate surface, resulting in narrower reflectance bandwidths, stronger reflection coefficients and better electro-optical properties [103]. However, the narrow temperature domain range of BPII and the small effect of friction treatment on the BPI alignment limit its practical application. Zheng et al. developed a facile method to achieve micropatterning of the crystal orientation of soft-standing BP superstructures by photo-aligning the substrate with a delicately designed photomask to display the alternating uniform and random orientation of the lattice crystal orientation at high resolution [104]. Erasure and rewriting can be stimulated by sequential UV irradiation and electric fields, as shown in Figure 11. The micropatterns can be erased by a combination of unpolarized UV irradiation and electric field stimulation, and then restored to the initial pattern by a combination of polarized UV irradiation and electric field stimulation of a differently shaped photo-mask. This work broadens the range of applications for blue-phase liquid crystals and provides an important guide to the controllability of crystal orientation in other soft organic and inorganic materials.

Recently, the study of soft coexistence systems of BPLC has attracted attention [105,106,107]. The lattice constants of soft cubic BPs (BPI and BPII) are determined by the pitch of the chiral LC, which is comparable to the wavelength of visible light, thus qualifying them as soft photonic crystals [20,108,109]. The stable coexistence of optically non-chiral anisotropic nematic-phase liquid crystals and optically chiral isotropic BPLCs were proposed and demonstrated by Mo et al. with the development of a local microregion polymer template to distribution adjusted to micropatterned techniques [110]. As shown in Figure 12, clear-defined micropatterns were obtained by photomask techniques and this soft-patterned coexistence system can greatly facilitate the understanding of the formation, arrangement and dynamics of soft condensed matter, thus promoting the development of various technological applications.

The simple cubic lattice of BPII is preferred over the body-centered cubic lattice of BPI because of its excellent stimulus responsiveness, operational compliance and more satisfactory photonic performance in applications [111,112]. Polyhedral oligomeric silsesquioxane has been shown to be advantageous in stabilizing BP [113,114]. Zhou et al. used the light-driven reversible transition of the simple cubic-BPII lattice to fabricate biphasic micropatterns containing both BPII and N* phases in two well-defined regions of a biphasic micropattern [115]. As shown in Figure 13, the photowritten pattern can be erased by UV irradiation and the pattern can subsequently be re-written by photolithography. This study demonstrates the photo-writing, UV erasure and rewriting of a dual-phase (BPII and N*) pattern and proves that this dual-phase micropattern is of considerable research value.

The optical writing patterned mode through masks can only produce prefabricated patterns, which is a complex process and does not allow for desirable patterned displays and does not meet the demand for patterned displays. In contrast, inkjet printing allows for the versatility and flexibility of BP patterned mode.

### 2.5. Maskless Inkjet Printing of Blue-Phase Patterned Mode

BP-based handwritten display devices are expected to replace conventional paper and protect the environment. However, BP handwriting films do not yet fully meet the requirements in terms of resolution control and pattern reproduction. Therefore, the realization of high-resolution BP patterns by inkjet printing has attracted a lot of attention. Inkjet printing technology is used extensively for printing text and images, and in recent years it has been used extensively in the biological and catalytic fields. Any high-resolution pattern can be prepared by inkjet printing, which could lead to major breakthroughs in the military, sensors, displays, etc.

#### 2.5.1. Humidity Responsive Inkjet Printing Blue-Phase Patterned Mode

The ability of photonic polymer coatings to adapt to their ever-changing surroundings has far-reaching implications for a variety of applications such as optical sensor devices, information concealment and environmental camouflage. Wang et al. prepared a humidity-responsive color-shifting photonic polymer layer based on hydrogen-bonded BPLC [116]. As shown in Figure 14, the BPLC polymer coating can be made to exhibit different color changes by adjusting RH, which is due to the selective expansion of the BPLC film resulting in a red shift of the color [6,117,118,119,120]. Inkjet printing of patterns can be achieved using this humidity-responsive film, and erasure of the pattern can be achieved by evaporating the water, thus enabling rewritability. This kind of photonic film has attracted a great deal of interest in both military and civilian applications as it enables BPLC patterning applications such as messaging and sensors, helping to camouflage and hide things from the environmental background.

#### 2.5.2. Solvent-Responsive Inkjet Printing of Blue-Phase Patterned Mode

Currently, pattern reproduction based on BP materials is only possible in non-cross-linked BPs, and most BP patterning applications are limited to two colors. As most BP materials are usually preserved in cells, which hinders their potential applications, there are still some challenges in fabricating BP polymer networks with stable panchromatic patterns [91,121]. Yang et al. prepared a printable photonic polymer coating using a single-domain BPLC network [122]. Areas printed with 5CB (liquid crystal) immediately swelled, resulting in an increase in BP pitch in the vertical direction and a color change from blue to red, and were reusable by erasing with tetrahydrofuran. The print resolution determines the ultimate functionality of the printed pattern in potential applications [123,124]. Meng et al. proposed a novel high-resolution “live” BPLC pattern based on simple control of ink diffusion on hydrophobically modified BPLC [125]. As shown in Figure 15, inkjet printing of liquid crystal 5CB on a modified BPLC film achieves high-resolution patterning while allowing erasure by N,N-dimethylformamide. Over time, the color of the printed pattern changes from green to cyan, and the pattern can be preserved on the film for a long time. This mode, which requires no response molecules, is easy to prepare and combines with inkjet printing, is of great significance for the development of advanced optical instruments.

## 3. Conclusions and Outlook

With the widespread use of blue-phase materials in display and non-display applications, research into blue-phase materials has become more and more popular. The blue phase has more obvious characteristics suited to pattern reproduction applications and therefore provides a wide range of application prospects. This paper reviews the application of BP micro-spontaneous composition, electric-field-driven BPLCD, handwriting-driven BPLC, photomask and ink-jet printing technologies over the years. The applications of BP pattern reproduction in each mode still have many challenges. Blue-phase spontaneous micropatterning through temperature changes and thus pattern changes can be applied to high-level anti-counterfeiting and exhibits good application prospects, but the stability is poor. BPLCD driven by electric fields has the advantages of fast response and easy adjustment, and is considered to be the basis of the next generation of LCDs, but it needs to overcome the problem of high driving voltage, etc. BPLC handwriting film has the advantages of easy writing and energy savings, and is an ideal replacement for traditional paper, but the low resolution and the complicated preparation process remain the problems for mass production at present. The patterning applications based on the mask and inkjet printing mode have the advantages of high resolution and wide application range, which can be applied to military information transmission, anti-counterfeiting, camouflage, etc. However, the problems of the complex substrate processing and frequent nozzle blockage have not been solved yet. More research is needed to bring the pattern reproduction applications of BPLC into people’s lives. This paper provides insights and relevant references for the research and development of new BPLC optical devices. Future research into BP pattern reproduction applications needs to focus on reducing the driving voltage, comprehensive development of multiple response modes through organic/inorganic composites and nanoparticle doping, etc. to simplify the process and achieve multifunctional applications.

## Figures and Tables

**Figure 1 materials-16-00194-f001:**
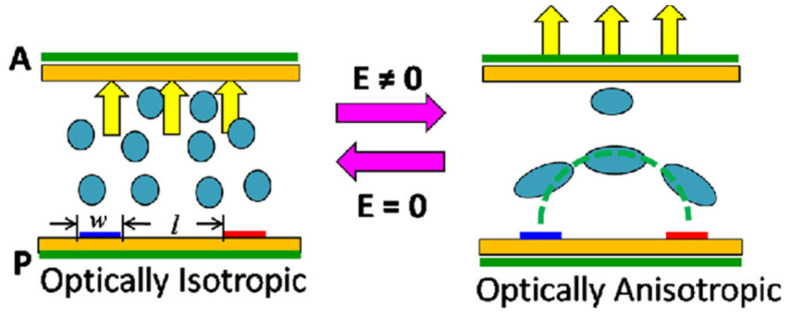
Schematic representation of the Kerr effect for blue-phase liquid crystals [50]. Emerging blue-phase LCDs from L Rao, Spienewsroom; published by Society of Photo-optical Instrumentation Engineers.

**Figure 2 materials-16-00194-f002:**
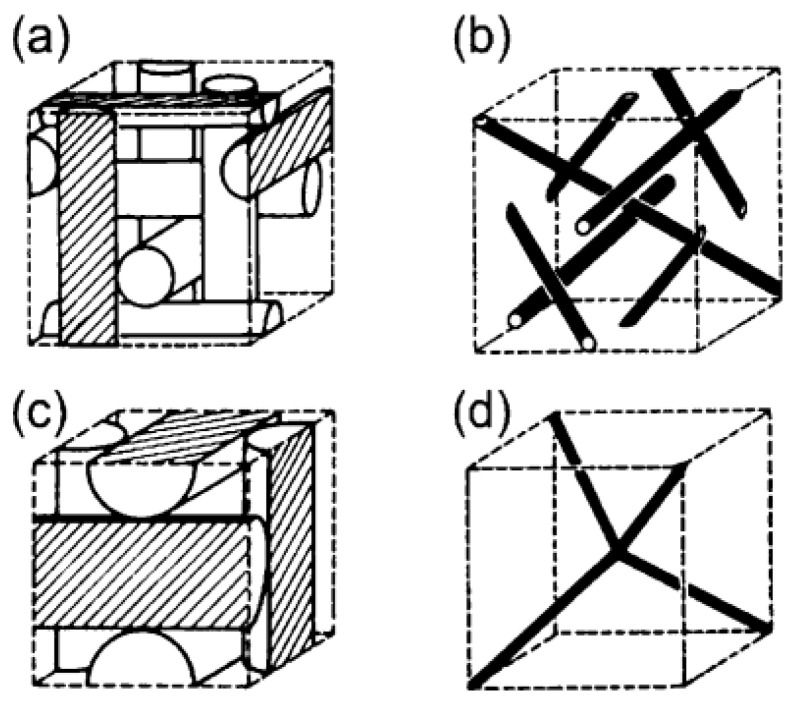
Structures of Blue Phases I and II. The rods in (**a**,**c**) represent the double-twist cylinder. The black lines in (**b**,**d**) represent disclination lines [59]. Liquid Crystalline Blue Phases from Kikuchi, Liquid Crystalline Functional Assemblies and Their Supramolecular Structures; published by Springer Nature.

**Figure 3 materials-16-00194-f003:**
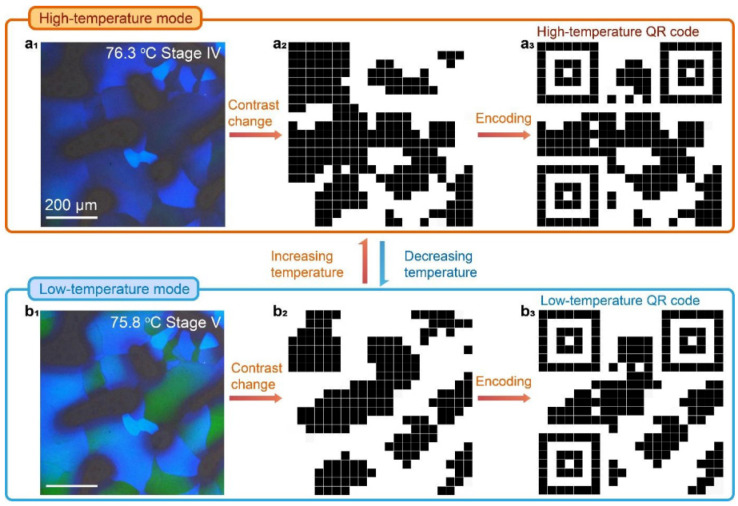
Temperature-controlled BP microscopic QR code transformation [62]. Diffusionless transformation of soft cubic superstructure from amorphous to simple cubic and body-centered cubic phases from L Jiang, Nat Commun; published by Springer Nature.

**Figure 4 materials-16-00194-f004:**
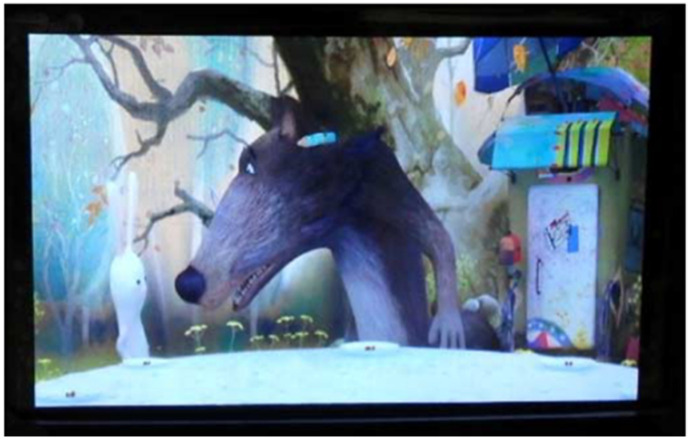
15-inch blue-phase liquid-crystal display [63]. Invited Paper: The World’s First Blue Phase Liquid Crystal Display from Lee, SID Symposium Digest of Technical Papers; published by John Wiley and Sons.

**Figure 5 materials-16-00194-f005:**
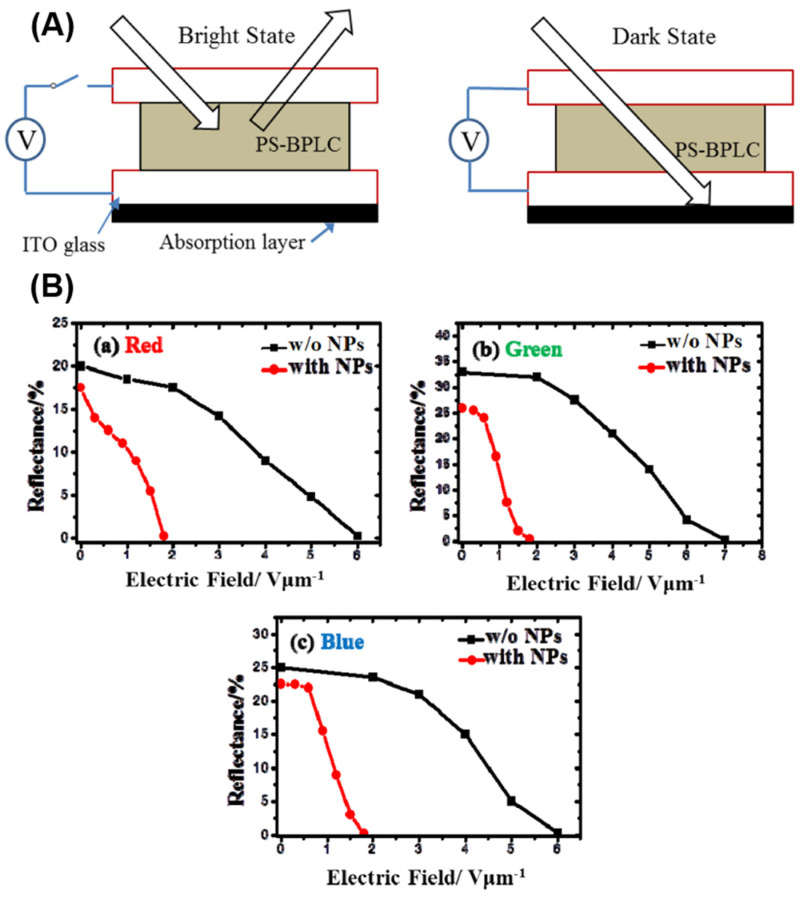
(**A**) Working principle of the proposed reflective display, (**B**) relationships of reflectance and driving electric field with and without doping BaTiO_3_ ferroelectric nanoparticle [71]. Low voltage polymer-stabilized blue phase liquid crystal reflective display by doping ferroelectric nanoparticles from Luo, Opt Express; published by OPTICAL SOC AMER.

**Figure 6 materials-16-00194-f006:**
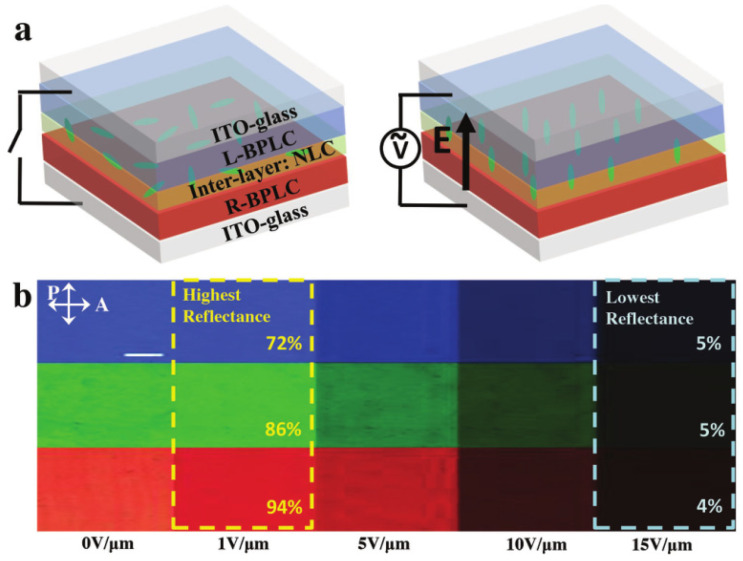
(**a**) Configurations of the BPLC film with and without an electric field. (**b**) Polarized optical microscopy images of the multi-layer BPLC film in blue, green and red color regions under increasing electric field [72]. Electrically Switchable, Hyper-Reflective Blue Phase Liquid Crystals Films from Luo, Advanced Optical Materials; published by John Wiley and Sons.

**Figure 7 materials-16-00194-f007:**
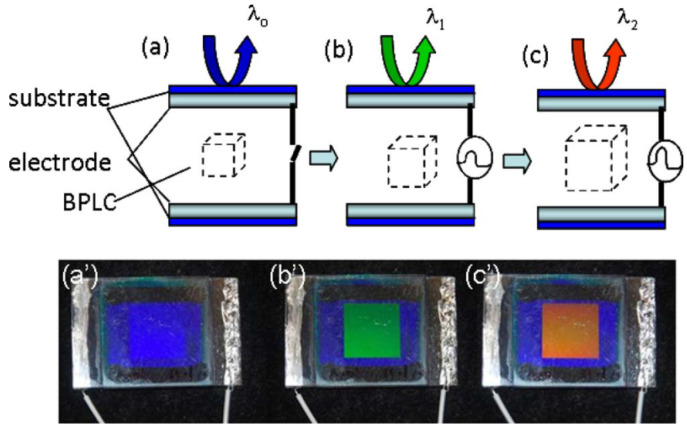
l Schematic illustration of electrically switched color of a PSBP cell in which the blue reflected wavelength (**a**) is switched to reflect a green color (**b**) and red (**c**) with an increase in applied voltage. Corresponding photographs of a one-pixel PSBP cell with 10μm cell gap is operated in the reflective mode with appearance of color by Bragg reflection under (**a’**) 0 V (blue), (**b’**) 33 V (green), and (**c’**) 40 V (red), respectively [73]. Electrically switched color with polymer-stabilized blue-phase liquid crystals from Lu, Optics Letters; published by OPTICAL SOC AMER.

**Figure 8 materials-16-00194-f008:**
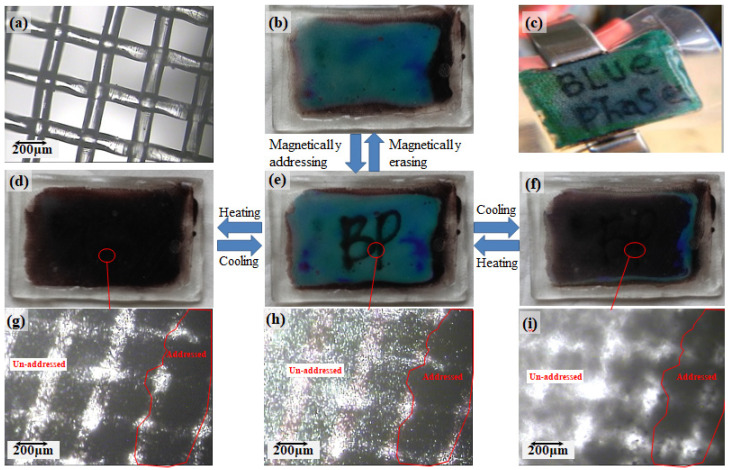
The “write-erase” process for magnetically driven BPLCDs. (**a**) Optical microscopy image of a nylon polymer network used in the LC composites; (**b**) photograph of LC cell with Fe_3_O_4_ doped BPLC before magnetically addressed; (**c**) photograph of Fe_3_O_4_ doped BPLC sandwiched between two flexible PET films; (**d**–**f**) photographs of the prepared cell containing Fe_3_O_4_ doped BPLC in the nylon polymer network after being magnetically addressed in isotropic phase, BP phase and N* phase, respectively; (**g**–**i**) optical microscopy image of the prepared cell containing Fe_3_O_4_ doped BPLC in the nylon polymer network after magnetically addressing in isotropic phase, BP phase and N* phase, respectively. [86]. Preparation and optical properties of Fe_3_O_4_ nanoparticles-doped blue phase liquid crystal from He, Phys Chem Chem Phys, published by ROYAL SOCIETY OF CHEMISTRY.

**Figure 9 materials-16-00194-f009:**
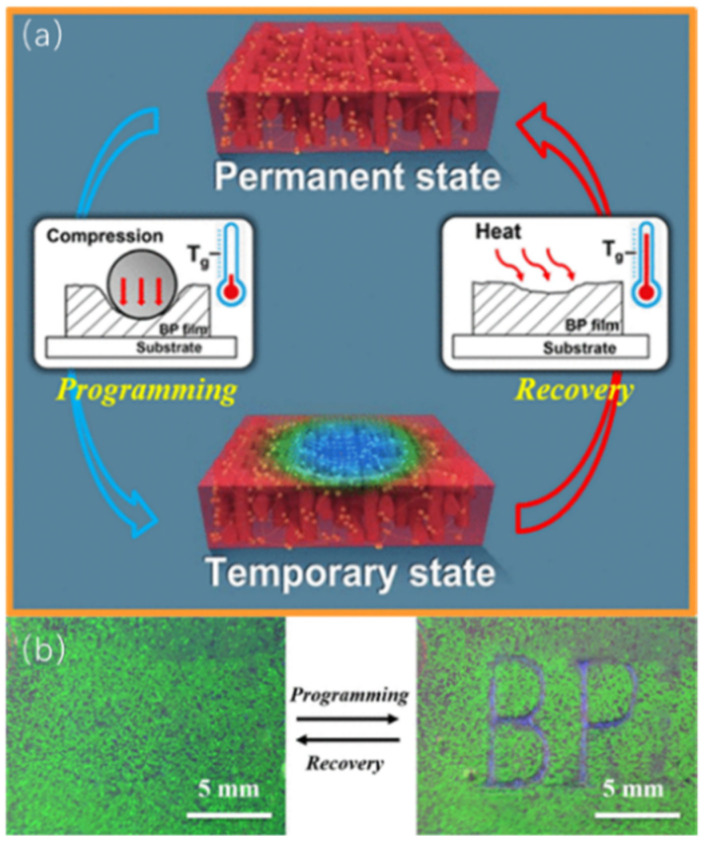
(**a**) Schematic diagram of the shape-memory process; (**b**) Writing and erasing of the BP pattern [90]. Photonic Shape Memory Polymer Based on Liquid Crystalline Blue Phase Films from Yang, Appl Mater Interfaces, published by American Chemical Society.

**Figure 10 materials-16-00194-f010:**
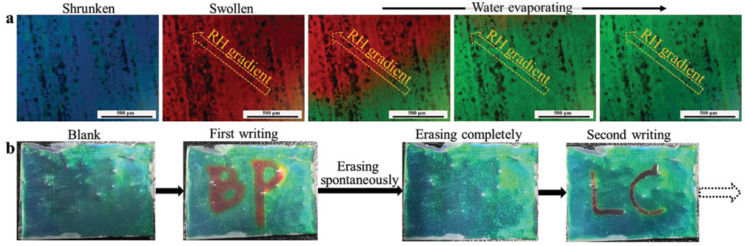
POM images of writing tablet model, photographs showing the responsive behaviors. (**a**) POM images of the color changing with the RH. (**b**) Photographs of the writing and erasing performance of the film [92]. Humidity-Responsive Blue Phase Liquid-Crystalline Film with Reconfigurable and Tailored Visual Signals from Yang, Advanced Functional Materials, published by John Wiley and Sons.

**Figure 11 materials-16-00194-f011:**
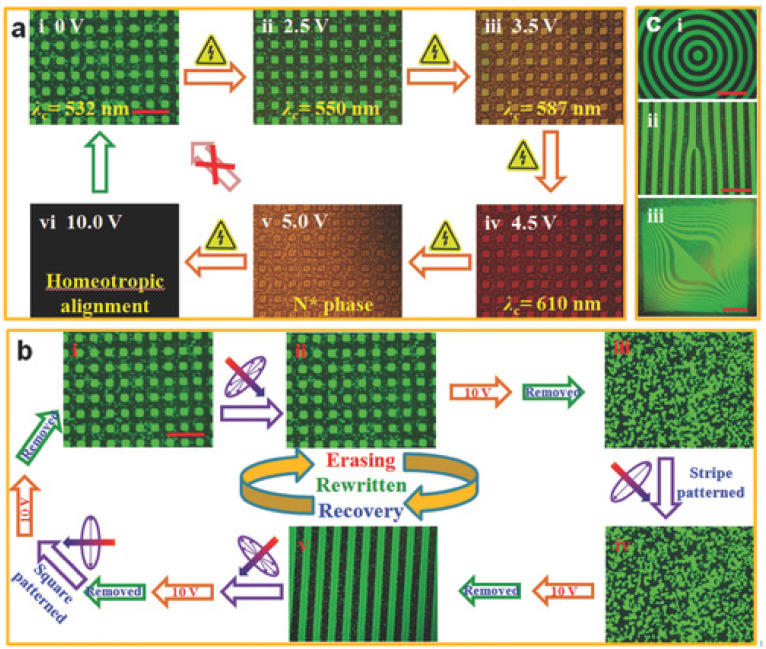
Stimulus orientation behavior of crystal orientation patterns formed by alternating uniform and random BP domains. (**a**) The BP lattice can be deformed by an applied voltage, leading to a red-shifting of the central wavelength of reflection band. (**b**) The pattern can be erased and rewritten by sequential UV-irradiation and electric-field stimulation. (**c**) Another different regular periodic patterns and irregular topological patterns can be rewritten [104]. Light-Patterned Crystallographic Direction of a Self-Organized 3D Soft Photonic Crystal from Zheng, Adv Mater, published by John Wiley and Sons.

**Figure 12 materials-16-00194-f012:**
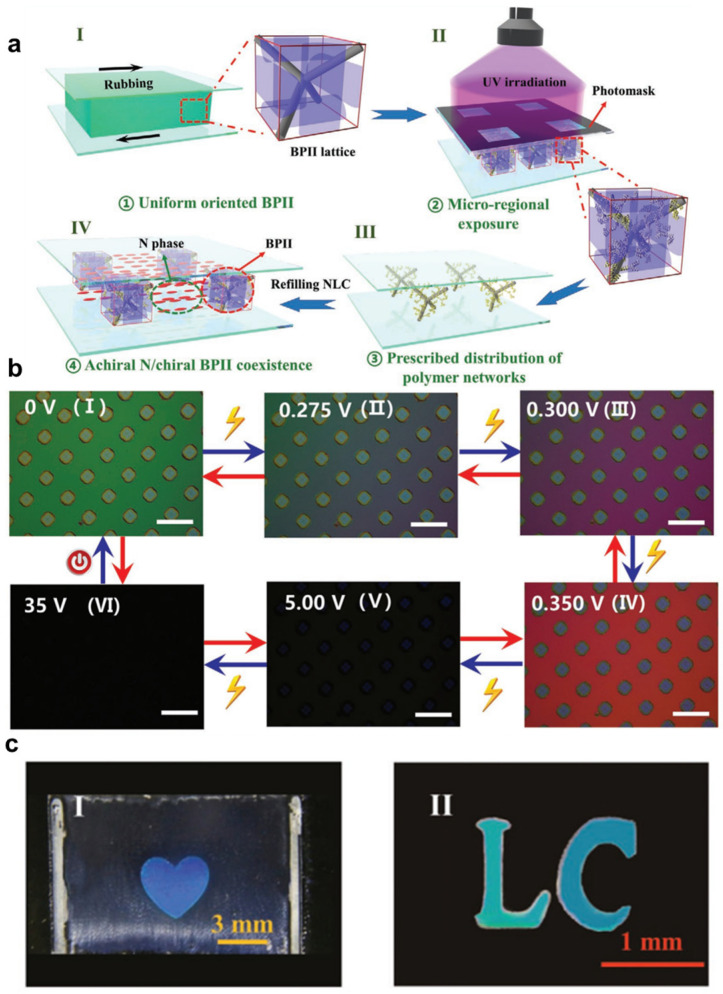
(**a**) Schematic diagram of the patterned soft coexistence of optically non-chiral anisotropic N phase and optically chiral isotropic BPII by local micro-regional polymer template technique; (**b**) reflected POM image of two phases coexisting at different applied voltages; (**c**) reflected POM image of two phases coexisting [110]. Reversible On–Off of Chirality and Anisotropy in Patterned Coexistence of Achiral-Anisotropic and Chiral-Isotropic Soft Materials from Mo, Advanced Optical Materials, published by John Wiley and Sons.

**Figure 13 materials-16-00194-f013:**
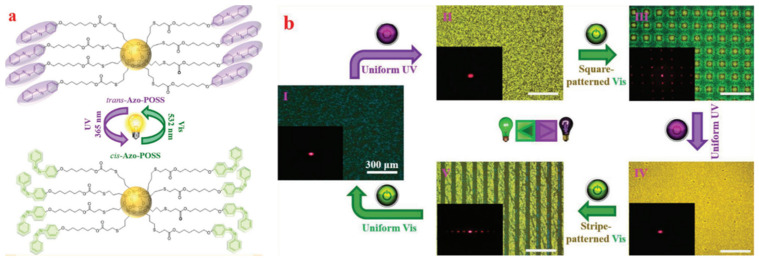
(**a**) Light-driven reversible transition between the self-organized BPII soft simple cubic lattice and the helical superstructure. (**b**) Fabrication of biphasic micropatterns containing BPII and N* under alternating UV and visible light irradiation [115]. Light-Driven Reversible Transformation between Self-Organized Simple Cubic Lattice and Helical Superstructure Enabled by a Molecular Switch Functionalized Nanocage from Zhou, Adv Mater, published by John Wiley and Sons.

**Figure 14 materials-16-00194-f014:**
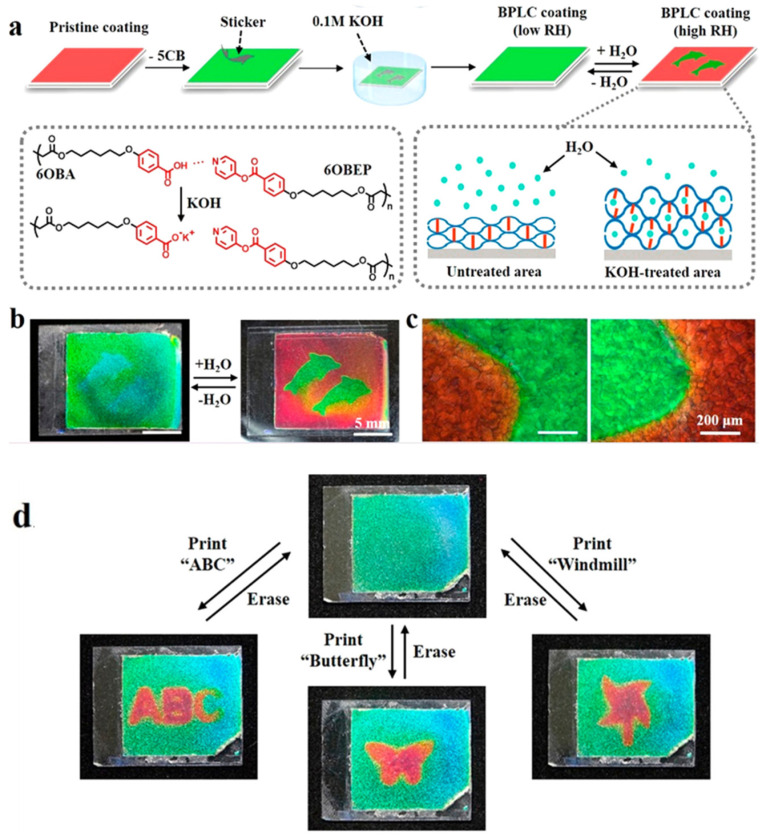
(**a**) Schematic diagram of BPLC pattern information encryption. (**b**) Input and erasure of pattern information. (**c**) POM images of humidity-driven BPLC film. (**d**) Printing and erasure of inkjet printing patterns [116]. Bioinspired Color-Changing Photonic Polymer Coatings Based on Three-Dimensional Blue Phase Liquid Crystal Networks from Wang, Appl Mater Interfaces, published by American Chemical Society.

**Figure 15 materials-16-00194-f015:**
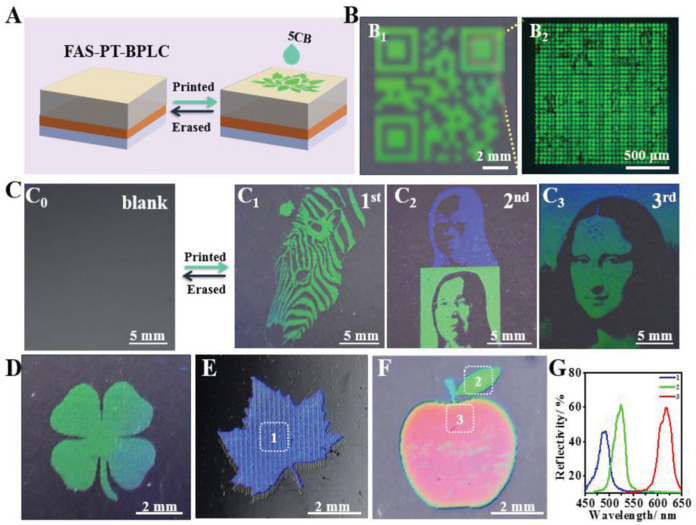
Various erasable patterns on BPLC membrane. (**A**) Scheme for the fabrication process of the erasable image on BPLC. (**B**–**F**) Photographs of the erasable images of the BPLC membranes. (**B**) Photo and reflective POM image of Quick code. (**C**) Reversible writing/erasing process of (C1) zebra, (C2) a double-color pattern with a portrait was printed using the image and inversed image as the pattern and (C3) Mona Lisa. The pattern was written and erased on the same substrate. Patterns of a (**D**) clover and (**E**) maple leaf, and (**F**) red apple with a green leaf and (**G**) the spectra of the samples with different regions in (**E**,**F**) [125]. High-Resolution Erasable “Live” Patterns Based on Controllable Ink Diffusion on the 3D Blue-Phase Liquid Crystal Networks from Meng, Advanced Functional Materials, published by John Wiley and Sons.

## Data Availability

Not applicable.

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
