# Peer review of "Research Progress on Blue-Phase Liquid Crystals for Pattern Replication Applications"

_materials, 2022, doi:10.3390/ma16010194_

Round 1

Reviewer 1 Report

The authors have given a comprehensive review of blue-phase crystals and their applications so far. However, I have a few concerns:

1. Sections 2.3 and 2.4 need to include more examples from the literature.

2. Authors should add the challenges and limitations of different kinds of configurations and applications of BPCs. 

3. A better resolution image of Figure 14 should be added.

Reviewer 2 Report

The manuscript attempts to cover the applications of liquid crystals that undergo a transition into a blue phase. The field is interesting, but the paper has several drawbacks. The language and style are problematic. The authors mix plural with singular (Blue phase liquid crystal(BPLC)[1-3] are fascinating, The applications of liquid crystals in patterned applications has always been a focus of attention). Furthermore, there are many four-line sentences and in some cases, they are not even complete! (In the future, the reduction of the driving voltage of electrically driven patterns through organic/inorganic composites, nanoparticle doping, etc., the integrated development of multi-response patterns to reduce the complexity of the preparation process as well as the realization of multi-functional applications.) Sometimes the authors did not bother to start a new sentence with a capital letter. The bibliographic research is inadequate and if this wasn’t enough many references are examined in groups, leading to a limited overview of the area. Many abbreviations are not defined for instance SID (page 3 Line 9), PSPBLC, PSBP, PBG.  Figure 1 is not referred to the text. In general, there is sloppiness throughout the whole work. In my opinion, it is not up to the standards of the journal. Many more work hours are needed.

Reviewer 3 Report

The manuscript entitled <Research Progress on Blue Phase Liquid Crystals for Patterned Applications> is a well-prepared review article on BP LCs targeted for specific applications. The work is novel and original. The introduction section clearly describes the state-of-the-art in the specific field of research which is well substantiated by the relevant references including the recent ones. The main recent advances on BP LCs are well- presented in this review. This review will definitely have high impact on the specific field of research and should be interesting especially for the readers dealing with BP-related materials and novel BP-based applications. The manuscript should be considered for publication at Materials-MDPI after a minor revision as indicated. Here are the specific comments:

(A) A minor revision of the language style by the native speaker might be necessary; this will increase the overall clarity of the review.

(B) Some important references are missing in this review: (i) on  blue phase photonic crystals (DOI: 10.1038/s41598-020-67083-6); (ii) on fast self-assembly of macroscopic blue phase 3D photonic crystals (DOI:  10.1364/OE.393197); (iii) on stabilizing of liquid crystalline blue phases (DOI: 10.1039/C2SM07155J). Those references are highly relevant to the reiwe..

(C) Used wording <beautiful> is non-informative and might not be used in the research articles. Please try to use another wording.

(D) In case if the figure used in the manuscript has been already published before somewhere, the proper reference should be used also in the respective figure caption. Please correct everywhere.

(E) Figure 11 is too small and hardly readable; please increase the size of this figure.

Reviewer 4 Report

Please see the attached pdf.

Reviewer 5 Report

Dear Authors

Your work is very interesting. However, it could be improved with suggestions given in the attached version of your manuscript. The Introduction part could be improved by adding new references. Also, there is a missing part in this topic like liquid crystal flakes that could be used in flexible reflective paper-like displays, this is just an example, and it is given in the suggestions within the manuscript.

The Main part of reviewed cases in the manuscript is fine and well organized. Please consider giving more details in Figures in terms of their titles and references. 

The conclusion part also needs improvement in terms of English editing and a better overview of the reviewed work, or what is the perspective of these materials in terms of a variety of applications.

I hope that these comments will contribute to a better manuscript.

Thank you.

Round 2

Reviewer 2 Report

The authors made substantial effort to improve the quality of the manuscript. Taking  additionally into account the opinion of the other reviewers  I agree to the publication of the work  

Reviewer 5 Report

Dear Authors

The manuscript is an improved version now. 

Thanks for improving and rewriting it in a better and more clarified way.

Some minor suggestions are given in the attached version.

Regards.
